# Acute Kidney Injury in Rhabdomyolysis: A 5-Year Children’s Hospital Network Study

**DOI:** 10.3390/healthcare12171717

**Published:** 2024-08-28

**Authors:** Jamie M. Pinto, Gregory Ison, Lora J. Kasselman, Srividya Naganathan

**Affiliations:** 1Jersey Shore University Medical Center, 1945 Route 33, Neptune, NJ 07753, USA; jamie.pinto@hmhn.org (J.M.P.); gregory.ison@hmhn.org (G.I.); 2Hackensack Meridian School of Medicine, 123 Metro Boulevard, Nutley, NJ 07110, USA; 3Hackensack Meridian Health Research Institute, 111 Ideation Way, Nutley, NJ 07110, USA; lora.kasselman@hmhn.org

**Keywords:** rhabdomyolysis, acute kidney injury, pediatric

## Abstract

Rhabdomyolysis is a skeletal muscle injury that can cause myoglobinuria and acute kidney injury (AKI). Risk factors for AKI in children are not clearly understood with no standardized treatment guidelines for rhabdomyolysis. Our study explores factors associated with AKI and management of pediatric patients with rhabdomyolysis. Medical records from a children’s hospital network over a 5-year period were retrospectively reviewed. The results are described with respect to the presence or absence of AKI. Of the 112 patients who met the inclusion criteria, AKI incidence was 7.1% (n = 8), with all affected patients having exertional etiology. The overall mean age was 13.5 years; patients without AKI were younger than patients with AKI (13.3 versus 17; *p* < 0.001). Using regression models for hypothesis generation, we found that patients with AKI were more likely to be older (OR = 1.44, 95%CI [1.11–2.19]; *p* = 0.03), have myoglobinuria (OR = 22.98, 95%CI [2.05–432.48]; *p* = 0.02), and have received intravenous bicarbonate (OR = 16.02, 95%CI [1.44–228.69]; *p* = 0.03). In our study, AKI was uncommon and associated with older age, myoglobinuria and bicarbonate treatment. Larger, prospective studies are needed to further understand AKI risk factors and optimal management of pediatric rhabdomyolysis.

## 1. Introduction

Rhabdomyolysis is a clinical condition characterized by skeletal muscle injury with the leakage of cellular breakdown products into the circulatory system [1,2,3]. Creatinine kinase (CK) levels greater than five times the upper limit of normal, or absolute levels greater than 1000 IU/L, are used for the diagnosis of rhabdomyolysis [2,4,5,6,7]. Viral infections, especially influenza, are among the most common etiologies of rhabdomyolysis in children [3,4,5,8] with recent recognition of SARS-CoV-2 as a causative agent [9]. Other etiologies described for pediatric patients include trauma, heat-related illness, severe exertion and metabolic disorders [2,4,8,10]. Clinically, rhabdomyolysis may present with a triad of myalgia, weakness and discolored urine [2].

Acute kidney injury (AKI) is a major complication of rhabdomyolysis. AKI develops as a result of decreased renal perfusion, glomerular cast formation and myoglobin induced oxidative tubular injury [2,4,5,7]. The reported incidence of AKI in pediatric rhabdomyolysis is highly variable and ranges from 5 to 45% [3,4,5,6,8,10,11,12,13,14,15]. Risk factors for the development of AKI are not clearly understood, with prior studies showing inconsistent association with age, sex, median peak CK values and urinary abnormalities [4,5,6,11,12,13].

The goal of management of rhabdomyolysis is to prevent the development of AKI or progression of renal deterioration if AKI is present [2]. Proposed management strategies include rigorous intravenous hydration and use of bicarbonate administration in fluids for alkaline diuresis [2,6]. Unfortunately, there is no consensus or standardized guidelines with respect to the type and amount of fluid selected, rate of administration or the addition of bicarbonate leading to variability in practice. The aim of the present study is to explore the incidence of and factors associated with AKI in pediatric patients with rhabdomyolysis, who presented to an acute care facility in our regional, pediatric network.

## 2. Materials and Methods

### 2.1. Study Population and Design

This was a retrospective chart review of patients selected from our hospital networks’ electronic health record using International Classification of Diseases (ICD)-10 codes for viral myositis M60.009, myositis M60.9, rhabdomyolysis M62.82 and myoglobinuria R82 during the study period, 1 January 2016 to 30 November 2021. Our regional healthcare network includes 3 inpatient pediatric units, spanning northern and central New Jersey. This study received approval from the Institutional Review Board of Hackensack Meridian Health (ID: Pro2021-1443).

### 2.2. Definitions

For the purposes of our study, we defined:Rhabdomyolysis as CK > 1000 IU/L [3,5,7];Myoglobinuria as patients with myoglobin in the urine and/or with urinalysis that was positive for blood but negative for red blood cells [16];AKI using Kidney Disease: Improving Global Outcomes (KDIGO) guidelines [17]. When baseline creatinine levels were unavailable, creatinine levels ≥ 0.3 mg/dL of the upper limit for age within 48 h of symptom onset were considered indicative of AKI;Maintenance fluid rate as calculated by the Holliday–Segar technique [18].

### 2.3. Inclusion Criteria

All pediatric patients from birth to 21 years who (1) met the definition for rhabdomyolysis and (2) presented for acute care to a pediatric hospital within our health network were included in our study. 

### 2.4. Exclusion Criteria

Patients were excluded if they did not meet criteria for rhabdomyolysis (coding error), they were previously known to have AKI or CKD due to other etiologies, or they had underlying metabolic muscle disorders or muscular dystrophies that predispose to rhabdomyolysis.

### 2.5. Statistical Methods

Descriptive statistics (number, frequency, mean, median, standard deviation, minimum, and maximum, as appropriate) were calculated for all variables overall and with respect to AKI. A multivariable logistic regression was used to assess associations between AKI and demographic factors (age and sex) as well as clinical factors (peak CK, myoglobinuria and IV bicarbonate). Variables were chosen based on associations identified in prior studies [4,5,6,11,12,13]. Other variables were not included in the final model due to autocorrelation, or clinical reasoning. Significance was set at *p* ≤ 0.05. All analyses were conducted using R version 4.3.3 (R Core Team, Vienna, Austria, 2021).

## 3. Results

Of 189 patients identified with ICD-10 codes, 112 were included in the data analysis. Of the patients excluded, 73 did not meet the definition of rhabdomyolysis and four had underlying muscle disorders. AKI was identified in 7.1% (n = 8) of patients. Overall, patients’ mean age was 13.5 years with patients in the AKI group being older (17 vs. 13.3 years; *p* < 0.001). The majority of patients were male (79.5%) regardless of group and almost all had no baseline creatinine values documented in the electronic medical record.

Demographic and clinical characteristics such as sex, race/ethnicity, insurance type and location of muscle pain were similar between the study groups (Table 1). Overall, exertional muscle use (59.8%), viral infections (32.1%) and other causes (8.1%) accounted for etiologies of rhabdomyolysis within our hospital network, with all cases of AKI having exertional etiology. Median peak CK values were higher in patients with no AKI when compared to patients in the AKI group (5710 IU/L vs. 2040 IU/L; *p* = 0.016). Myoglobinuria was present in 37.5% of patients with and 24% of patients without AKI (Table 1). 

Management and outcomes are described for patients with and without AKI (Table 2). Patients in the AKI group received more fluid boluses (median 2 vs. 1.5) as well higher rates of continuous fluid administration when compared to calculated maintenance rates (median 2 vs. 1.5), although these differences were not significant. A total of 26 patients (23.2%) received sodium bicarbonate either as a bolus infusion (n = 2) or administered with continuous fluids (n = 24). By group, 37.5% of patients with and 22.1% of patients without AKI received any bicarbonate. The majority, regardless of group (n = 21; 81%), received ½ Normal Saline with added sodium bicarbonate, ranging from 25 to 77 mEq/L, administered as a continuous infusion. Length of stay (LOS) for patients with and without AKI was 28.5 h and 56.5 h, respectively (*p* = 0.064). During acute hospitalization, none of our AKI patients required hemodialysis and most (63%) normalized creatinine prior to discharge. Of the three patients with elevated creatinine at discharge, two were lost to follow-up and one had persistent elevation in creatinine consistent with chronic kidney disease.

With the logistic regression analysis, patients with AKI were more likely to be older (OR = 1.44, 95%CI [1.11–2.19]; *p* = 0.03), have myoglobinuria (OR = 22.98, 95%CI [2.05–432.48]; *p* = 0.02), and to have been administered IV bicarbonate (OR = 16.02, 95%CI [1.44–228.69]; *p* = 0.03) (Table 3). There was no difference between groups by sex or peak CK values.

## 4. Discussion

In this exploratory study, within our regional healthcare network, we identified 112 patients with rhabdomyolysis in a 5-year period, of whom 7.1% had AKI and the majority were male. Using descriptive statistics, patients with AKI were found to be older and to have lower peak CK levels. Hypothesis testing with regression analysis demonstrated an association of older age, myoglobinuria and administration of IV bicarbonate with AKI.

The reported incidence of pediatric AKI is variable with ranges from 5 to 45% [3,4,5,6,10,11,13,14], mostly in studies that were small and single-center [3,4,5,6,10,11,13,14], some of which did not exclude patients with underlying muscle disorders [5,15]. A larger database study, which utilized the Kids’ Inpatient Database and included 8599 patients from the United States, reported an incidence of AKI of 8.5%, which is similar to our study finding [12].

Prior studies to identify factors that predict or are associated with AKI in pediatric rhabdomyolysis have reported inconsistent results. Most studies do not show any association of a specific etiology of rhabdomyolysis with the development of AKI [3,4,10,11,19] unless an underlying muscle disorder was present [5,15]. In contrast, although not statistically significant, all patients with AKI in our study had exertional etiology as the cause of rhabdomyolysis, which may be related to the high rate of exertional etiology in the overall sample. Akin to our research findings, several studies report an association of AKI with older age [4,11,12,15]. However, other studies showed no association with age [3,19] or an association with younger age [14]. Additionally, while prior studies with larger sample sizes have correlated male sex with AKI [4,12,15], the present report and others with smaller numbers of enrolled patients found no association of AKI with sex [3,19].

The median peak CK level in our study is 4890 IU/L, similar to prior reports [4,10,20]. We found no correlation between peak CK level and AKI, which is consistent with previous studies [3,4,11,12,19] and, interestingly, patients in our sample without AKI had higher median CK levels than patients with kidney injury. Similarly, Gardner et al. showed an inverse correlation of median peak CK levels with AKI [4] in a single-center, United States study of 319 patients with rhabdomyolysis secondary to viral myositis. Their study found that patients without AKI had median peak CK levels of 4000 IU/L compared to 2400 IU/L in patients with AKI [4]. In contrast, multiple studies have demonstrated elevated CK levels in association with AKI [5,8,10,13,14,15], albeit with different cut-off levels ranging from IU/L to 20,000 IU/L [5,8,10,13,15] and several with reported differences failing to meet statistical significance [8,10].

Abnormal urinary findings, such as the presence of blood and myoglobin were well-described in children with rhabdomyolysis and AKI [5,8,10,13,15]. Similar to our study, previous reports [10,13,15] demonstrated a positive correlation between myoglobinuria and the development of AKI in pediatric patients. Other studies [5,8] reported an association of AKI with positive blood, which can indirectly indicate myoglobin [16]. Interestingly, in a study by Gardner et al., AKI was not associated with the presence of myoglobinuria. Overall, given myoglobin’s short half-life and rapid clearance from circulation, it is an unreliable biomarker with high false negative rates [2,21].

There are few published reports investigating treatment strategies and outcomes for pediatric rhabdomyolysis. Few studies report the use of bicarbonate in intravenous fluids for children with rhabdomyolysis and a diagnosis of AKI [5,10]. Like the present report, Lim et al. found in their retrospective analysis of 39 Korean children that bicarbonate was more often administered to patients with AKI than to children with no kidney involvement [10]. Conversely, in a study by Mannix et al. of 210 pediatric patients from a United States hospital, there was no association of elevated creatine levels with bicarbonate therapy, but there was a correlation with higher fluid rates [5]. Given the paucity of data and variable results from the existing literature, larger and prospective studies are needed to assess the effects of treatment strategies on outcomes.

### Limitations

Our study has several limitations that are important to note. First, this was a retrospective, exploratory study from a single hospital network; therefore, the results may not be representative of the general pediatric population. Second, patients were identified using ICD-10 codes; therefore, we may have missed patients due to misclassification. Additionally, we did not have baseline creatinine levels for most patients, urine output data or consistent documentation of height with which to use the Schwartz formula to estimate baseline creatinine [4]. Although this is a scenario encountered commonly in clinical practice, estimating baseline creatine at the upper limit of age for normal likely means we underestimated the true incidence of AKI in our study population. Finally, our sample contained only eight patients with AKI which resulted in wide confidence intervals on regression analysis. However, since this was an exploratory study, it identifies potentially important variables associated with AKI and provides a foundation for further research. Future studies that include more patients with AKI could help address this limitation.

## 5. Conclusions

Our study describes demographic, clinical and treatment factors for pediatric patients with rhabdomyolysis. AKI, which was not common in our study population, was associated with older age, myoglobinuria and treatment with IV bicarbonate. Patients with AKI did not have higher peak CK values when compared to patients without AKI. Larger and more prospective studies are needed to explore and understand the risk factors for AKI and the optimal treatment strategies for pediatric rhabdomyolysis.

## Figures and Tables

**Table 1 healthcare-12-01717-t001:** Demographic and clinical characteristics of pediatric patients with rhabdomyolysis with respect to Acute Kidney Injury.

	AKI	No AKI	Overall	*p*-Value
	(n = 8)	(n = 104)	(n = 112)	
Age (years)				<0.001
Mean (SD)	17.0 (1.85)	13.3 (5.47)	13.5 (5.38)	
Sex				0.999
Male	7 (87.5%)	82 (78.8%)	89 (79.5%)	
Female	1 (12.5%)	22 (21.2%)	23 (20.5%)	
Race/Ethnicity				0.999
Non-Hispanic White	4 (50.0%)	52 (50.0%)	56 (50.0%)	
Non-Hispanic Black	1 (12.5%)	14 (13.5%)	15 (13.4%)	
Hispanic	2 (25.0%)	26 (25.0%)	28 (25.0%)	
Other	1 (12.5%)	12 (11.5%)	13 (11.6%)	
Insurance				0.829
Public, Charity or None	2 (25.0%)	45 (43.3%)	47 (42.0%)	
Private	6 (75.0%)	59 (56.7%)	65 (58.0%)	
Location of Pain				0.568
Lower extremity	4 (50.0%)	43 (41.3%)	47 (42.0%)	
Upper extremity	1 (12.5%)	21 (20.2%)	22 (19.6%)	
Other or multiple locations	3 (37.5%)	40 (38.5%)	43 (38.4%)	
Etiology				0.129
Exertional	8 (100%)	59 (56.7%)	67 (59.8%)	
Viral	0 (0%)	36 (34.6%)	36 (32.1%)	
Other *	0 (0%)	9 (8.7%)	9 (8.0%)	
Peak Creatine Kinase (IU/L)				0.016
Median [Min, Max]	2040 [1060, 32,000]	5710 [1040, 242,000]	4890 [1040, 242,000]	
Maximum creatinine (mg/dL)				0.002
Median [Min, Max]	1.40 [1.00, 2.39]	0.765 [0.250, 1.70]	0.800 [0.250, 2.39]	
Maximum creatinine/reference value				0.131
Median [Min, Max]	1.75 [1.34, 1.93]	0.650 [0.240, 1.34]	0.685 [0.240, 1.93]	
Myoglobinuria				0.41
Present	3 (37.5%)	25 (24.0%)	28 (25.0%)	
Absent	5 (62.5%)	79 (76.0%)	84 (75.0%)	

* Other etiologies include trauma, heat, medication and vaccine side effects.

**Table 2 healthcare-12-01717-t002:** Management and outcomes of pediatric patients with rhabdomyolysis with respect to Acute Kidney Injury (AKI).

	AKI	No AKI	Overall	*p*-Value
	(n = 8)	(n = 104)	(n = 112)	
Total number of fluid boluses				0.454
Median [Min, Max]	2.00 [1.00, 3.00]	1.50 [0, 4.00]	1.63 [0, 4.00]	
Maintenance fluid type				0.029
Normal saline	3 (37.5%)	79 (76.0%)	82 (73.2%)	
Lactated Ringer’s solution	0 (0%)	2 (1.9%)	2 (1.8%)	
Mixed fluids *	2 (25%)	4 (3.8%)	6 (5.4%)	
No continuous fluids prescribed	3 (37.5%)	19 (18.3%)	22 (19.6%)	
Fluid rate/calculated maintenance rate				0.468
Median [Min, Max]	2.00 [1.50, 2.00]	1.50 [1.00, 4.25]	1.50 [1.00, 4.25]	
Bicarbonate administration				0.376
Present	3 (37.5%)	23 (22.1%)	26 (23.2%)	
Absent	5 (62.5%)	81 (77.9%)	86 (76.8%)	
Length of Stay (h)				0.064
Mean (SD)	28.5 (35.2)	56.5 (51.1)	54.5 (50.5)	

* Any combination of normal saline and Lactated Ringer’s solution.

**Table 3 healthcare-12-01717-t003:** Logistic regression: factors associated with Acute Kidney Injury (AKI).

Characteristic	OR	95%CI	*p*-Value
Age (years)	1.44	1.11–2.19	0.03
Sex			
Male	—	—	—
Female	0.89	0.02–13.65	0.94
Peak Creatine Kinase (IU/L)	1.00	0.99–1.00	—
Bicarbonate administration			
Absent	—	—	—
Present	16.02	1.44–228.69	0.03
Myoglobinuria			
Absent	—	—	—
Present	22.98	2.05–432.48	0.02

## Data Availability

Deidentified data are stored at the corresponding author’s institution and can be obtained via written request.

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
