# Peer review of "Acute Kidney Injury in Rhabdomyolysis: A 5-Year Children’s Hospital Network Study"

_healthcare, 2024, doi:10.3390/healthcare12171717_

Round 1

Reviewer 1 Report

Comments and Suggestions for Authors

This study is well designed and topic of study is very important and interested. Neverthless authors should calculate a power analyse and sample size. Due to number of AKI (n=8) is too small to make a statistical analysis. 

This kind of study should include more AKI patients to have a true analysis. 

Reviewer 2 Report

Comments and Suggestions for Authors

I hope the editor and authors find the attached file.

Reviewer 3 Report

Comments and Suggestions for Authors

The manuscript provides valuable insights into the association of acute kidney injury (AKI) with various factors in pediatric patients with rhabdomyolysis. I have some comments, especially in the abstract section:

1-      I recommend you remove this part form the abstract:

“Continuous variables are summarized with means, standard deviations 16 (SD), medians and ranges, and categorical variables with counts and percentages. Regression analysis is presented as odds ratio (OR) with 95% confidence interval (95%CI)”.

2-      I recommend you add the number of patients: “Of 112 patients who met 18 inclusion criteria, AKI incidence was (n = 8) 7.1%. All affected patients had exertional etiology”

3-       I recommend you insert the mean age as well.

General:

-            Explain the reason behind the very wide confidence interval in myoglobinuria and IV Bicarb.

-            Address why your study found an association between exertional etiology and AKI while others did not.

-            The finding that male sex was not associated with AKI in your study, unlike some prior studies, warrants further discussion. Consider exploring potential reasons.

Round 2

Reviewer 1 Report

Comments and Suggestions for Authors

The validity of these results can be enhanced by conducting future studies with an increased number of patients. The most important point that convinced me to acept for publication is that the patient group is valuable.

Reviewer 2 Report

Comments and Suggestions for Authors

The reviewer thinks the authors responded and revised the manuscript appropriately. 

Reviewer 3 Report

Comments and Suggestions for Authors

No further comments